# Divide and Conquer Self-Supervised Learning for High-Content Imaging

## Abstract

Self-supervised representation learning methods often fail to learn subtle or complex features, which can be dominated by simpler patterns which are much easier to learn. This limitation is particularly problematic in applications to science and engineering, as complex features can be critical for discovery and analysis. To address this, we introduce Split Component Embedding Registration (**SpliCER**), a novel architecture which splits the image into sections and distils information from each section to guide the model to learn more subtle and complex features without compromising on simpler features. SpliCER is compatible with any self-supervised loss function and can be integrated into existing methods without modification. The primary contributions of this work are as follows: i) we demonstrate that existing self-supervised methods can learn shortcut solutions when simple and complex features are both present; ii) we introduce a novel self-supervised training method, SpliCER, to overcome the limitations of existing methods, and achieve significant downstream performance improvements; iii) we demonstrate the effectiveness of SpliCER in state-of-the-art medical and geospatial imaging settings. SpliCER offers a powerful new tool for representation learning, enabling models to uncover complex features which could be overlooked by other methods.

## 1 Introduction

Recent advances in acquiring highly detailed, information-rich imaging at scale have resulted in high-content imaging – imaging which aims to maximise data capture – being widely used in science (Way et al., 2023), medicine (Radtke et al., 2022; Lin et al., 2023), and engineering (Xia et al., 2010). In general, however, computer vision methods are designed for natural images such as ImageNet (Deng et al., 2009), which can be well-described with relatively simple features (Singla & Feizi, 2021). High-content imaging generally contains subtle, complex features which can be difficult to distinguish from noise without strong supervision. The effect of this can be that simple features dominate the representation of the image, and subtler features are ignored. However, subtle or complex features can be among the most interesting features for scientific discovery, such as the interaction or morphology of subtly different cell types (Hale et al., 2024), as simpler features are generally more readily discovered by humans.

In many cases, we can leverage an intuitive reason or some prior knowledge indicating that the features of a region or channel of an image are of particular interest. It is usually possible to split the image into smaller components, perhaps by generating a segmentation mask or cropping the image, or by separating channels in multiplex images. This information can be used to guide the model to learn features from each component (Farndale et al., 2023a;b; 2024; Nakhli et al., 2024). Ensuring features are distributed across all components could ensure that complex features which cluster in a certain component will be learned. However, we generally lack the methods to enable this type of *divide and conquer* strategy for high-content imaging.

A prime example is multiplex imaging: information-dense images containing more than three channels, each with a distinct meaning and significant variation in the complexity of the features in each channel. Despite their prevalence, multiplex images are generally understudied in computer vision, and largely lack dedicated methods to extract key information from them.

In this work we introduce a novel representation learning architecture, *Split Component Embedding Registration (SpliCER)*, which deconstructs an image into components, and distils information from

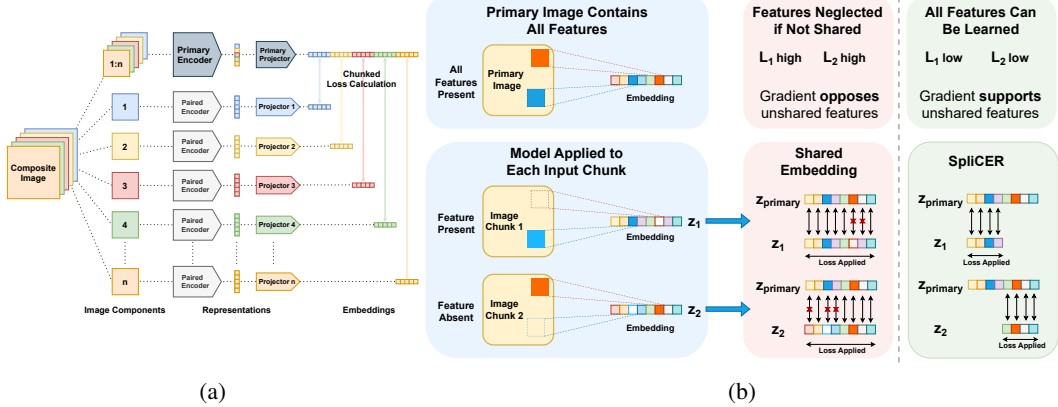

(a)                                               (b)

Figure 1: (a) SpliCER training architecture. (b) Schematic of the benefits of SpliCER compared to mapping multiple inputs to a shared latent space.

each component to direct models to learn features from each part, significantly improving representation learning and downstream performance. This forces the model to learn features from each component, meaning components with more complex or subtler features cannot be ignored. As this deconstruction is only performed during training, the encoder can then be used downstream without requiring deconstruction. SpliCER is able to utilise any self-supervised architecture or loss, so it is generally applicable across self-supervised methods.

## 2 BACKGROUND AND RELATED WORK

Self-supervised learning (SSL) uses inherent structure in data to learn useful representations without relying on manual labels. This is achieved using a proxy objective which directs the model to learn features satisfying certain conditions (Shwartz-Ziv & LeCun, 2023), without prior knowledge of the desired downstream task(s). Recent work demonstrates that many of the most effective methods map multiple views of the same input into a *joint embedding* (JE) (Giakoumoglou & Stathaki, 2024), maximising the mutual information shared between views. Typically, this is achieved through augmenting the input to artificially create different views, with augmentations designed to avoid augmenting the key semantic features which the model should learn (Bachman et al., 2019).

Consequently, models have a tendency towards learning simple features, as these are generally more robust to augmentations (Jing et al., 2021). However, as models have no signal with which to determine the true importance of features, this can lead models to exhibit *simplicity bias* and ignore more informative, complex features (Shah et al., 2020; Pinto et al., 2022; Vasudeva et al., 2023). Models must compress their inputs, and the structural properties used can often favour *shortcut solutions*, where models learn redundant features which are correlated with informative features in the training set (Geirhos et al., 2020; Robinson et al., 2021; Chen et al., 2021). This can lead models to learn low-dimensional representations of the true underlying manifold, missing key predictive features that would be desirable for downstream tasks. Complexity should not be confused for density (e.g. Wang et al. (2021)), which refers to the spatial coverage of representations learned for every local region of an image, as opposed to the semantic intricacy of those representations.

There have been efforts to address these shortcomings by leveraging prior knowledge about the data. By aligning multiple inputs in a shared latent space, complex features which may be ignored in a unimodal setting may be learned if they align well with simple features in the paired inputs (Radford et al., 2021; Girdhar et al., 2023; Farndale et al., 2023a). However, it has been demonstrated that alignment in a shared latent space can reduce performance if predictive information is present in only one modality (Farndale et al., 2023b). This is because the model loses its implicit supervisory signal for all features which are not present in the paired information. For example, it is impossible to completely describe the semantics of an image with words, so a vision/language model will have a considerable amount of the content of the image not shared between inputs. Similarly, an image

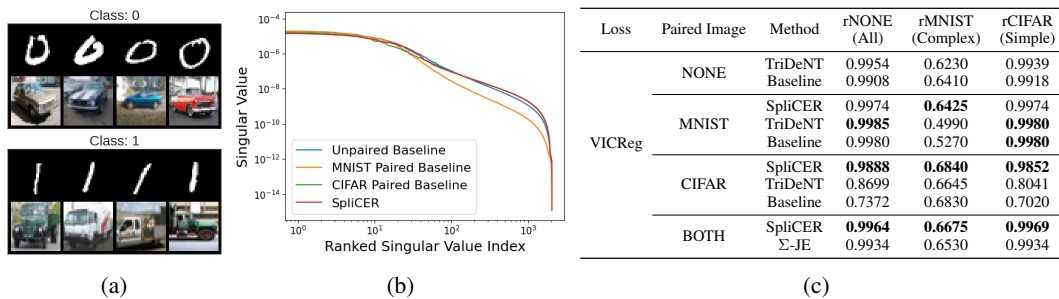

| Loss | Paired Image | Method | rNONE (All) | rMNIST (Complex) | rCIFAR (Simple) |
|------|--------------|--------|-------------|------------------|-----------------|
| | NONE | TriDeNT | 0.9954 | 0.6230 | 0.9939 |
| | | Baseline | 0.9908 | 0.6410 | 0.9918 |
| | MNIST | SpliCER | 0.9974 | **0.6425** | 0.9974 |
| VICReg | | TriDeNT | **0.9985** | 0.4990 | **0.9980** |
| | | Baseline | 0.9980 | 0.5270 | **0.9980** |
| | CIFAR | SpliCER | **0.9888** | **0.6840** | **0.9852** |
| | | TriDeNT | 0.8699 | 0.6645 | 0.8041 |
| | | Baseline | 0.7372 | 0.6830 | 0.7020 |
| | BOTH | SpliCER | **0.9964** | **0.6675** | **0.9969** |
| | | Σ-JE | 0.9934 | 0.6530 | 0.9934 |

(a)                    (b)                              (c)

Figure 2: (a) MNIST-CIFAR examples (b) Ranked singular values of models trained on MNIST-CIFAR (c) SSL performance for models trained and evaluated on randomised MNIST (rMNIST), randomised CIFAR (rCIFAR), or with no randomisation (rNONE).

could be equally well-described with poetry or a technical description. Neglecting this unshared information represents a major limitation of methods such as CLIP (Radford et al., 2021).

*TriDeNT* (Farndale et al., 2023b) addresses these issues by using multiple joint-embeddings to enable model to receive signal from both unimodal and multimodal objectives, but it is limited by having to align entire representations across modalities. While potentially leading to emergent alignment (Girdhar et al., 2023), it has been shown that this can overconstrain models, meaning features not shared between branches are ignored, especially as more input modalities are added. These architectures are typically motivated by using a paired input to improve the performance of a model for a *primary* input. We refer to the encoder of this primary input as the *primary encoder*, and the encoder of the paired input as the *paired encoder*.

## 2.1 MULTIPLEX IMAGING

Widely used in domains as varied as medicine (Kobayashi et al., 2010), biology (Lewis et al., 2021), and geoscience (Drusch et al., 2012), *multiplex/multichannel* images are *hyperstacks* of spatially registered greyscale images each containing different information about the same scene or object. We consider two types of multiplex imaging: *spatial proteomics* and *hyperspectral imaging*.

**Spatial proteomics** are a range of methods used to analyse the spatial distribution of proteins in tissue samples, winning Nature Methods' Method of the Year 2024 (Nature Methods, 2024). These techniques generate multiplexed images where each channel represents the localisation of a specific protein, revolutionising researchers' capacity to understand the morphology and interactions of different cell types. However, manual analysis of these images requires extensive knowledge of each protein and its typical distribution. Typical analysis reduces each cell to a vector of mean expression in each channel (Brbić et al., 2022; Shaban et al., 2024), neglecting morphology, secreted proteins, and extracellular matrix components (Bussi & Keren, 2024). There is a paucity of manually labelled datasets, meaning that supervised approaches have limited efficacy, and can only reproduce known phenotypes. Images from different experiments usually feature different numbers of channels with different proteins, prohibiting the use of generalist foundation models (Vorontsov et al., 2024; Xu et al., 2024).

**Hyperspectral imaging** captures frequencies outside the range of human vision in 13 bands. The electromagnetic spectrum contains rich information beyond visible light that can be utilised for applications from food processing (Gowen et al., 2007) to urban planning (Weber et al., 2018) and atmospheric monitoring (Stuart et al., 2019). A key application is land-use classification, where hyperspectral bands from satellite images contain information such as cloud cover, vegetation health, geology, and soil moisture content (Drusch et al., 2012). These images are rapidly produced at scale, with the Sentinel-2 satellites covering the entire surface of the earth approximately every five days (Drusch et al., 2012). Machine learning tools which can fully leverage the vast amounts of data generated by these satellites could discover critical information on climate change and land use patterns (Prexl & Schmitt, 2023). However, like medical imaging, different bands have significantly different feature complexity, meaning critical subtle or complex information may be ignored.

## 3 SPLICER: SPLIT COMPONENT EMBEDDING REGISTRATION

Despite the difficulty in creating manually labelled datasets for supervised learning, it can be relatively simple to systematically isolate the image component containing a feature of interest. This could be done manually, using some prior knowledge about the image structure, particularly for multiplex images which are naturally decomposed. Alternatively, this could be automated with a model such as *Segment Anything* (Kirillov et al., 2023). Methods which can leverage this information without overconstraining the model by enforcing pairwise embedding alignment will be able to train models which identify complex features in images and achieve better downstream performance.

We introduce *Split Component Embedding Registration (SpliCER)*, a self-supervised training architecture which allows models to flexibly learn features from all components of a deconstructed image while avoiding shortcut solutions. SpliCER optimises this process by i) mapping the primary (not deconstructed) image to an embedding, ii) mapping each component of the image to a distinct embedding, iii) splitting the primary image's learned embedding into *chunks*, with the embedding of each component registered to a distinct chunk of the primary embedding.

This creates a separate joint embedding between the embedding of each deconstructed input component and a chunk of the primary image's embedding. Figure 1b illustrates the intuition behind this. When the embeddings of all branches must be aligned to the entire primary embedding, this necessitates that branches which do not contain a feature must align their embedding to branches with that feature. These embeddings therefore contain random noise or collapse to a constant, resulting in gradients which oppose learning that feature in the primary embedding. In contrast, SpliCER aligns only a portion of the primary embedding to each branch's embedding, resulting in gradients not opposing features which are only present in one or few branches.

Consider a primary input $\bar{x}$, with paired inputs $x_1^*, \ldots, x_n^*$, a primary representation $\bar{z}$, and paired representations $z_1^*, \ldots, z_n^*$, mapped into embeddings $\bar{e} \in \mathbb{R}^N$ and $e_1^*, \ldots, e_n^* \in \mathbb{R}^M$ respectively. Splitting $\bar{e}$ into $n$ chunks such that $\bar{e} = \bar{e}_1 || \bar{e}_2 || \ldots || \bar{e}_n$, we can construct an optimisation problem

$$\mathcal{L} = \sum_{i=1,\ldots,n} \mathcal{L}_i(\lambda_i) = \sum_{i=1,\ldots,n} I(x_i; e_i | \bar{x}) + I(\bar{x}; \bar{e} | x_i) - \lambda_i \left[ I(\bar{x}; e_i) - I(x_i; \bar{e}) \right] \quad (1)$$

following Equation S4. We assume that all chunks are of equal size for brevity. Note that different chunks of the embedding are optimised separately, meaning that there is less risk to the model predicting low-variance features from the paired inputs, as these do not come at the cost of features from a different paired input, and there is no need for different chunks to be aligned in embedding space. Note also that this formulation removes the requirement that all inputs are jointly optimised, as each branch is only optimised relative to a chunk of the primary branch. As the primary embedding $\bar{e}$ will, in general, have a significantly greater dimension than the primary representation $\bar{z}$, an optimal model should condense features shared between different chunks into few elements of $\bar{z}$, allowing more complex features to be learned.

## 4 EXPERIMENTS

As there are few established methods for our problem setting, we first consider some preliminary elements motivating the design of SpliCER and justify why these form strong baselines. The first baseline we use in each experiment is a standard self-supervised architecture, either VICReg (Bardes et al., 2022) or SimCLR (Chen et al., 2020) (denoted *Baseline*). We use ResNet (He et al., 2016) models for each input, with the number of channels determined by the number of image channels. Full details are provided in Section S3.

We also include mapping each input into a shared latent space as a baseline (denoted $\Sigma$-JE), to study the aggregation of different joint embeddings. The input to the primary branch is the original image, while the input to each additional branch is a component of the image. This allows us to isolate the impact of the *chunking* aspect of SpliCER, compared to the impact of simply aligning inputs in a latent space. Where applicable we also include TriDeNT, however, there are few scenarios considered where there is only one source of paired data, as TriDeNT requires.

We provide full descriptions of the datasets and training hyperparameters used in Sections S2 and S3 respectively. We also provide model ablations in Section S4.

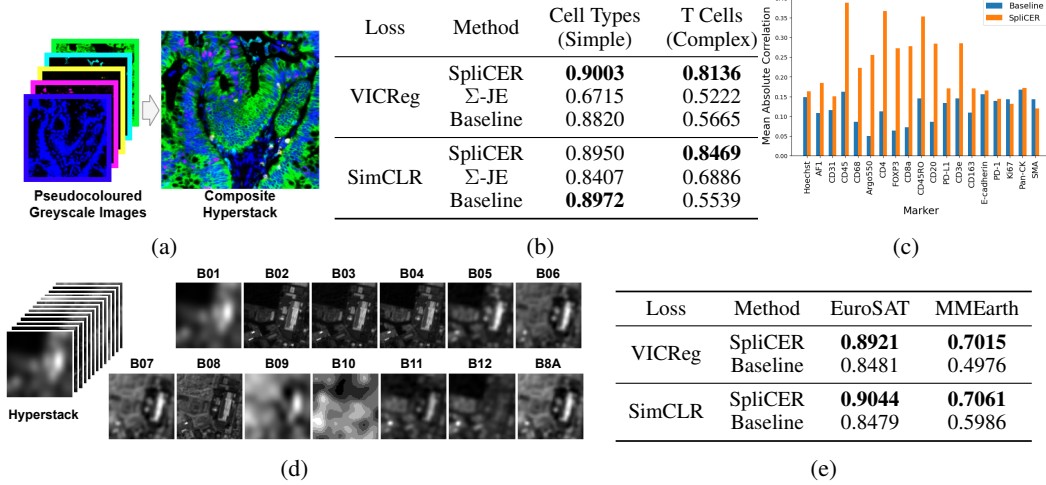

Figure 3: (a) Example of a multiplex immunofluorescence hyperstack (b) Orion-CRC classification accuracy (c) Mean absolute correlation between representation elements and marker intensities on Orion-CRC for baseline and SpliCER (d) Example of a hyperspectral image hyperstack (e) Hyperspectral imaging classification accuracy

**MNIST-CIFAR Simplicity Bias Evaluation.** Our first experiment compares the complexity of features learned by SpliCER to those learned by baselines. We use the MNIST-CIFAR dataset, an established benchmark for simplicity bias, to assess whether the model learns *simple* features, *complex* features, or both. The dataset is an amalgam of the MNIST handwritten digit dataset (LeCun et al., 1998), which contains simple features, and the CIFAR-10 dataset, which contains more complex features. Each image in the MNIST-CIFAR dataset is constructed by concatenating an MNIST image to a CIFAR image with the same label, such that the model concurrently processes both. Classes are mapped such that 0 is always car and 1 is always truck, so a model can find shortcut solutions by only learning MNIST features, as these are just as informative about the label as the CIFAR features, but are easier to learn.

We evaluate which features have been learned by training a classifier while randomising either the MNIST or CIFAR images, denoted as rMNIST and rCIFAR respectively, with rNONE being unrandomised. On rMNIST, a model which has learned a complete shortcut solution will have accuracy around 50%, as it will not be able to make predictions from the CIFAR features. On rCIFAR, this is unlikely to have an effect on a model which has learned a shortcut solution. Models which have learned more complex features will be less affected by MNIST randomisation and more affected by CIFAR randomisation.

In Figure 2c we demonstrate that baseline SSL training does not learn a complete shortcut solution, with accuracy of 64% on rMNIST and near perfect (∼100%) performance on rNONE/rCIFAR. However, this indicates that predictive features have not been learned, as a model trained on the CIFAR data alone achieves 72.6% accuracy. Figure 2b demonstrates the effect of the choice of paired data on the learned features. Pairing MNIST results in a lower rank representation, while the model trained with CIFAR as paired data has similar rank to unpaired training and SpliCER. We pair either the MNIST image, the CIFAR image or both, to compare the efficacy of different distillation methods. We find that SpliCER can make up to a 4 percentage point improvement when given CIFAR as paired information. This implies that the signal from the CIFAR branch guides the model to learn more complex features.

In baseline or TriDeNT, using MNIST as paired information appears to cause the model to focus even more on the simple features and collapse to a shortcut solution, resulting in poorer performance on rMNIST, but marginally increasing performance on rCIFAR. Similarly, using CIFAR as paired information increases performance on rMNIST, but heavily degrades performance on rCIFAR. This is because the paired information forces the model to primarily pay attention to the shared features, at the expense of the unshared features.

| Loss | Method | Histology | | X-ray |
| | | NCT | Camelyon | PneumoniaMNIST |
|---|---|---|---|---|
| VICReg | SpliCER | **0.9372** | **0.8230** | **0.8718** |
| | Baseline | 0.8855 | 0.6822 | 0.8446 |
| SimCLR | SpliCER | **0.9368** | **0.8674** | **0.9119** |
| | Baseline | 0.9067 | 0.8346 | 0.8462 |

| (a) | (b) |
|---|---|

Figure 4: (a) Sample segmentations for ChestMNIST (top) and NCT (bottom) (b) Classification accuracy on NCT, Camelyon and PneumoniaMNIST tasks

SpliCER achieves similar performance to the baseline and TriDeNT on the task that corresponds to the paired information, but also completely mitigates the performance degradation observed in the other models. This implies that SpliCER can benefit from being guided by the paired data, but the chunking of its embedding does not restrict its ability to learn the remaining features. Achieving a classification accuracy of 68.4%, the SpliCER approaches the performance of a model trained on the CIFAR images alone, which achieves an accuracy of 72.6%, without the option to learn the simple features in MNIST.

**Spatial Proteomics.** We next demonstrate the applicability of SpliCER to multiplex immunofluorescence images. As there are no established benchmarks for multiplex immunofluorescence, we construct three datasets from the Orion-CRC dataset (Lin et al., 2023): a pretraining dataset subsampled from all cell types with a distribution reflecting normal tissue distribution; Cell Types - a balanced evaluation dataset for classifying different cell types which require relatively simple features; and T Cells - a more fine-grained balanced evaluation dataset for classifying subtypes of T cells, requiring more complex features. As discussed in Section 2.1, there are significant differences in the variance of features in different channels of multiplex image, as shown in Figure S1a. The Cell Types task requires models to learn both simple and complex features, while T Cells assesses the whether the model has learned features of the low variance markers CD4, CD8, and FOXP3.

Consistent with the MNIST-CIFAR tasks, we find that SpliCER significantly improves performance for learning the complex features without degrading performance on the simple features. This leads to a consistent performance of around 89% on the cell type classification task, and a large improvement from 57/55% to 81/85% on the T cell subtyping task for VICReg/SimCLR respectively. Furthermore, we show in Figure 3c that SpliCER learns features with a considerably greater correlation with the marker intensities of the inputs, confirming that SpliCER is able to learn features of the individual markers more effectively than standard self-supervised methods.

**Hyperspectral Geodata.** We next assess SpliCER's performance in analysing hyperspectral geodata. We pretrain on either the EuroSAT (Helber et al., 2019) or MMEarth (Nedungadi et al., 2024) datasets, which contain 13-band hyperspectral imaging from the Sentinel-2 satellite program. These bands each correspond to an interval of the electromagnetic spectrum, and each band contains distinct information about a different aspect of the environment, such as atmospheric conditions or soil moisture. We evaluate the quality of the learned representations on the EuroSAT land use classification task, which requires models to use information from all spectral bands to make effective classifications.

Figure 3e shows that the land-use classification accuracy is significantly improved from 85% to 89-90% by using SpliCER on the 13-band images. In contrast, restricting the model to only use the RGB channels demonstrates no significant different in performance between the methods, and a worse performance overall of 79.67% for SpliCER and 79.78% for the baseline VICReg model. This demonstrates that incorporating the additional bands improves classification accuracy, and that ensuring the model learns from all of these additional channels significantly improves performance compared to baseline SSL approaches. We see an even greater increase in performance for SpliCER above baseline when pretraining on MMEarth, with accuracy improved from 50/60% to 70/71% for VICReg/SimCLR respectively, indicating that learning more complex features makes models more robust to distribution shift.

## 4.1 SEGMENTATION MASKS AS IMAGE DECONSTRUCTION

Until now the images used for each task have been naturally predisposed to deconstruction due to their compositional design. This is not necessary for SpliCER to provide improved performance, however, as we can use segmentation masks to deconstruct images. In many cases, we have prior knowledge that features of a particular part of the image should be learned, and this can be leveraged with segmentation to improve the quality of the learned representations. By using a segmentation to deconstruct an image into a region of interest and its complement, the model is still able to learn any image features, is no longer able to neglect either segmented component.

**Histopathology Nuclei Segmentation.** It has been shown that histopathology models routinely ignore cell nuclei, instead learning simpler, higher-variance features related to the connective tissue (Farndale et al., 2023b; 2024). Here we use HoVer-Net (Graham et al., 2019) to generate nuclei segmentation masks for each image in the NCT-CRC-100K dataset (Kather et al., 2019), which we then use to generate images containing only nuclei, or only connective tissue, as shown in Figure 4a. SpliCER can then be used to distil information from both paired images, with the goal of learning features from the nuclei as well as the connective tissue. The simple features here are still useful for downstream prediction, so it is no longer desirable to disregard these. We use synthetically generated segmentation masks to demonstrate that there is no need for exhaustive manual annotation, particularly with the availability of generalist segmentation models such as *Segment Anything* (Kirillov et al., 2023). There is also a very real biological application of this task, as representation learning is increasingly being used for biological discovery in histopathology (Bahadir et al., 2024), but could be hampered by models only learning simple features.

We evaluate on the NCT tissue type classification and Camelyon metastasis detection (Bandi et al., 2018) tasks. Camelyon features a training set of images from three hospitals, and a test set from a different hospital. This results in a distribution shift due to the significant differences in the imaging artefacts between sets. To achieve better performance on Camelyon, models must learn biologically robust features that generalise beyond the training set. Figure 4 demonstrates that SpliCER outperforms baseline accuracy on both tasks: 94/94% on NCT – a 5 percentage point improvement over baseline, and 82/87% on Camelyon, compared to a baseline of 68/83% for VICReg/SimCLR respectively. We evidence in Table S2 that the background features are useful to learn for these tasks, as models paired with only the nuclei mask have worse downstream performance.

**X-Ray Lung Segmentation.** Chest radiographs are typically used to identify very subtle changes in the body, and require extensive training to use effectively. A slight change in brightness can indicate the presence of disease, which may be lost to standard self-supervised models. Radiologists will typically look at the expected site of disease to identify these changes, but the context of the surrounding tissue is also important for identifying disease. We use the HybridGNet model (Gaggion et al., 2022) to generate lung segmentation masks for the ChestMNIST (Yang et al., 2023) dataset. These are then used in SpliCER with to direct the model to complex features specifically in the lungs, which are the primary site of the disease. We evaluate the models' performance on PneumoniaMNIST, which is a binary classification task prediction the presence of pneumonia. SpliCER achieves a significantly improved accuracy, with 87/91% compared to the baseline of 84/85% for VICReg and SimCLR respectively.

## 5 ANALYSIS

We have established empirically that standard SSL approaches can ignore complex features in favour of learning only simpler features. Here we analyse this phenomenon formally to provide some intuition for its causes. Let $X = (X_1, \ldots, X_n)$ be a composite image consisting of components $X_1, \ldots, X_n$, where $X_k$ represents the $k$th component. These components represent image features, for example, in the MNIST-CIFAR example we have two components $X = (X_{\text{MNIST}}, X_{\text{CIFAR}})$. We have an encoder $f$ yielding a representation $z = f(X; \theta_f)$ parameterised by $\theta_f$ and a projector $g$ yielding the embedding $e = g(X; \theta_g)$ parameterised by $\theta_g$. For some SSL joint embedding loss $\mathcal{L}_{JE}(e', e'')$, for embeddings $e', e''$ of augmented views $X', X''$ of $X$. Considering the gradient $\frac{\partial \mathcal{L}_{JE}}{\partial e}$ on the encoder and projectors' parameters $\theta_f$ and $\theta_g$ respectively, we have

$$\frac{\partial \mathcal{L}_{JE}}{\partial \theta_f} = \frac{\partial \mathcal{L}_{JE}}{\partial e} \cdot \frac{\partial e}{\partial z} \cdot \frac{\partial z}{\partial \theta_f}, \qquad \frac{\partial \mathcal{L}_{JE}}{\partial \theta_g} = \frac{\partial \mathcal{L}_{JE}}{\partial e} \cdot \frac{\partial e}{\partial \theta_g}. \tag{2}$$

The term $\frac{\partial \mathcal{L}_{JE}}{\partial e}$ is a single vector dictating the sensitivity of the total loss to changes in the embedding. Suppose that component $X_k$ contains *simple* features which are highly effective for minimising $\mathcal{L}_{JE}$, which could be encoded by some embedding $e_k$. These features are likely to be quickly discovered and reinforced by successive early iterations, resulting in an embedding $e \approx e_k$ being learned. As this will significantly decrease the loss value, the the term $\frac{\partial \mathcal{L}_{JE}}{\partial e}$ will push the model to converge towards this state, compared to deviating away from $e_k$. Consequently, the gradients favouring other features are quickly diminished, as these can incur large penalties for deviating from the simple, learned features.

The critical issue is that there is no mechanism in standard SSL to push the model to improve the representation of distinct features, as this could incur a larger loss value. There is no guarantee that these features would lead to a lower optimal loss value, and requiring that they are learned may increase the optimal loss value if they are less predictable. Particularly in cases where the images are dense in features, a limited representation capacity would require models to s acrifice highly predictable, simple features to learn these more complex features. Unlike supervised learning, there is no intrinsic signal where the model could experience any added value from these features, meaning that they will be ignored.

We propose that this simplicity bias is driven by the information bottleneck induced by mutual information maximisation. Following the information bottleneck principle (Balestriero et al., 2023). From Equation S3, we have

$$I(X'; e') = \underbrace{I(X'; e'|X'')}_{\text{superfluous information}} + \underbrace{I(X''; e')}_{\text{predictive information}} . \tag{3}$$

The objective implicitly maximises the predictive information $I(X''; e')$ and minimises the superfluous information $I(X'; e'|X'')$ unique to view $X'$ given $X''$. If features from $X_k$ dominate the predictive information, a reasonable strategy for the model is to robustly learn these features at the expense of features $X_j$ which may be less consistently shared between views, or harder to identify, causing them to be viewed as 'superfluous'.

## 5.1 SpliCER Learns Complex Features by Decoupling Optimisation

SpliCER is explicitly designed to overcome these issues, with a loss that separately optimises learning features from each component. This prevents features from one component from dominating the learning process for chunks associated with other components, and enables the model to learn complex features which do not reduce the loss or even incur a higher loss than simple features. By associating each component $X_i^*$ (via its embedding $e_i^*$) with a unique, dedicated chunk $\bar{e}_i$ of the primary embedding $\bar{e}$, SpliCER creates largely decoupled optimization pathways for the features of different components within the primary encoder.

**Gradients with respect to the projector's parameters $\theta_g$.** The gradient of the total loss with respect to the projector's parameters $\theta_g$ is

$$\frac{\partial \mathcal{L}_{\text{SpliCER}}}{\partial \theta_g} = \sum_{j=1}^{n} \frac{\partial \mathcal{L}_{JE,j}(\bar{e}_j, e_j^*)}{\partial \theta_g} = \sum_{j=1}^{n} \frac{\partial \mathcal{L}_{JE,j}(\bar{e}_j, e_j^*)}{\partial \bar{e}_j} \cdot \frac{\partial \bar{e}_j}{\partial \theta_g}. \tag{4}$$

The term $\frac{\partial \mathcal{L}_{JE,j}(\bar{e}_j, e_j^*)}{\partial \bar{e}_j}$ is the gradient of the $j$th loss component with respect to the $j$th chunk of the primary embedding, and is determined solely by the relationship between $\bar{e}_j$ and $e_j^*$. There is no direct impact of any other chunk $\bar{e}_k$ ($j \neq k$), or any other component embedding $e_k^*$.

Suppose that we have a sufficiently sparse primary projector that the final layers can be partitioned such that a subset $\theta_{g,j} \subset \theta_g$ predominantly or exclusively produces the chunk $\bar{e}_j$. This will mean that $\frac{\partial \bar{e}_k}{\partial \theta_{g,j}} \approx 0$ for $j \neq k$. Therefore, the gradient for $\theta_{g_j}$ is

$$\frac{\partial \mathcal{L}_{\text{SpliCER}}}{\partial \theta_{g,j}} \approx \frac{\partial \mathcal{L}_{JE,j}(\bar{e}_j, e_j^*)}{\partial \theta_g} = \frac{\partial \mathcal{L}_{JE,j}(\bar{e}_j, e_j^*)}{\partial \bar{e}_j} \cdot \frac{\partial \bar{e}_j}{\partial \theta_g} \tag{5}$$

This shows that the parameters $\theta_{g,j}$ which produce the chunk $\bar{e}_j$ are updated based on the supervisory signal from $\mathcal{L}_{JE,j}$, which is specific to component $X_j^*$, decoupled from the other components.

**Gradients with respect to the encoder's parameters $\theta_f$.** Similar to above, the chain rule gives

$$\frac{\partial \mathcal{L}_{\text{SpliCER}}}{\partial \theta_f} = \sum_{j=1}^{n} \frac{\partial \mathcal{L}_{JE,j}(\bar{e}_j, e_j^*)}{\partial \theta_f} = \sum_{j=1}^{n} \frac{\partial \mathcal{L}_{JE,j}(\bar{e}_j, e_j^*)}{\partial \bar{e}_j} \cdot \frac{\partial \bar{e}_j}{\partial z_{\text{primary}}} \cdot \frac{\partial z_{\text{primary}}}{\partial \theta_f}. \tag{6}$$

The primary encoder produces the shared representation $z_{\text{primary}}$, which is used to derive all chunks $\bar{e}_j$, meaning $f_{\text{primary}}$ receives gradient contributions from all $n$ component-specific terms. Here we see that each gradient contribution is formed of a component-specific term $\frac{\partial \mathcal{L}_{JE,j}(\bar{e}_j, e_j^*)}{\partial \bar{e}_j}$, which is independent of all other terms. Therefore a large improvement in the representation of features from a specific component will be reinforced with a strong gradient signal, regardless of the behaviour of the other components. The primary encoder must produce a representation which is sufficiently expressive that the projector can form all chunks to satisfy their respective loss terms. If this is not achieved, a large loss will be incurred from the affected chunks. However, if a chunk has reached its optimal embedding while still incurring a large loss, its gradients will saturate and have less impact on the overall learning, allowing other chunks to be prioritised in the optimisation process. This way, even small improvements in each chunk can be beneficial.

This occurs irrespective of how well other components $X_k^*$ (for $k \neq j$) being represented or how low their corresponding losses $\mathcal{L}_{JE,k}$ are. In a standard SSL setup with a single, unchunked embedding, if features from $X_k^*$ (e.g., simple MNIST features) can already minimize the total loss effectively, the impetus to learn features from $X_j^*$ (e.g., complex CIFAR features) might be diminished or lost. SpliCER avoids this by ensuring that there is a dedicated loss component and a corresponding portion of the primary embedding whose quality is judged solely on its ability to represent $X_j^*$.

Therefore, the learning pressure for features from component $X_j^*$ (acting through $\mathcal{L}_{JE,j}$ and its influence on $\bar{e}_j$) is preserved and not overridden by the learning status of other components. This allows $f_{\text{primary}}$ to be trained to recognize and encode features from all components $X_1^*, \ldots, X_n^*$ into $z_{\text{primary}}$, as each has a distinct pathway to exert its learning influence.

## 6 DISCUSSION

Shortcut solutions are a key issue for self-supervised learning approaches. In the absence of a supervisory signal to enforce the learning of complex features, we have shown that models can default to learning shortcut solutions. We have shown that SpliCER integrates well with existing self-supervised losses, and is highly effective for training models to learn complex features, enabling many more settings to make use of available prior knowledge, particularly multiplex images and those with segmentation masks available. With the development of generalist segmentation models such as *Segment Anything* (Kirillov et al., 2023), there are a wide range of fields where SpliCER could be used.

SpliCER can be easily adapted to suit different use cases. If a particular channel is known to be more informative, then more features can be allocated to that channel, or if a channel is known to not be informative, then it can be allocated fewer features or omitted altogether. Different losses could be applied for each embedding chunk, allowing customisation of the learned features.

**Limitations.** Regardless of the pretraining method, the classifier head used for downstream tasks is still susceptible to shortcut solutions. If the encoder has simple and complex features but the simple features occur more frequently or are more predictive, these are the features which will be used by the classifier, even if there is value in the complex features. There has been considerable existing work on mitigating simplicity bias (Vasudeva et al., 2023; Tiwari & Shenoy, 2023). These techniques could be combined with SpliCER, potentially creating a complementary effect in further reducing simplicity bias.

SpliCER is also highly dependent on the quality of the image deconstruction. There may not always be an easy way to segment images, or we may lack the knowledge to do so. Additionally, SpliCER could in theory negatively impact features which span multiple components, as their signal may be impacted by being separated.

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
