## S1 FURTHER RELATED WORK

### S1.1 SIMPLICITY BIAS AND DOMAIN GENERALISATION

Representation learning relies on proxy metrics to determine which features should be learned. Typical methods seek to optimise desirable properties of the representation, such as maximising the mutual information between embeddings (Chen et al., 2020; Ozsoy et al., 2022), or minimising the covariance between features (Bardes et al., 2022). *Simplicity bias* (Shah et al., 2020) is the tendency of machine learning models to learn the simplest features required to satisfy any given objective (Robinson et al., 2021; Chen et al., 2021), deteriorating the quality of models' learned representations. Models must seek to reduce redundancy in their representations so learning every possible complex feature is not possible, as these could be mostly noise. However, in many cases, learning the simplest features is a *shortcut solution* that neglects the complexity of the true underlying manifold (Geirhos et al., 2020).

The encoder architecture has been shown to have little effect on this bias (Pinto et al., 2022). A natural avenue for development is to incorporate additional sources of information into training which make it easier to identify these features. Whilst not explicitly designed to solve simplicity bias, recent work has established the effectiveness of this method, such as vision-language modelling (Radford et al., 2021) and knowledge distillation from paired data (Farndale et al., 2023b). However, these additional sources of information may not always be available.

Shortcut solutions are often exposed with transfer learning, as these features typically do not generalise as well as solutions which capture the complexity of the underlying manifold (Vasudeva et al., 2023). Sometimes simpler features are desirable in this setting, as they are usually more robust to minor changes in appearance. However, many simple features such as texture or background can be artefacts of a training set and consequently are harmful to domain generalisation.

### S1.2 SELF-SUPERVISED LEARNING FEATURE COMPLEXITY

It has been shown that self-supervised models learn features with a variance greater than that imposed by the augmentation regime, as these are the only features which can be reliably predicted under augmentation (Jing et al., 2021). In this setting, features which either only exhibit small changes in appearance or have larger, infrequent changes, will typically be very hard to differentiate from noise induced by augmentation. This reduces overfitting on trivial or undesirable features, but also results in models failing to learn the potentially large space of features which are valuable but have low variance. Naturally, features can not be easily normalised to unit variance, as they are not known a priori. For example, these interesting low-variance features could be minor changes in shape which are hidden by elastic deformations induced by the augmentation regime such that the model does not learn to identify them. If there are few features with sufficient variance to be learned relative to the augmentation regime, this phenomenon can lead to dimensional collapse, where a model produces a low-rank representation (Jing et al., 2021) and deteriorates performance (Wang & Qi, 2022).

Sridharan & Kakade (2008) showed that retaining predictive features while discarding superfluous information requires a *multi-view assumption* about the relationship between the views and downstream labels. This is the assumption that for two SSL inputs $x_1, x_2$, the meaningful semantics which should be learned are shared by both inputs (views). Formally, we write that for two views $x_1, x_2$ and some task label $y$, there exists some $\epsilon_{\text{info}} > 0$ such that

$$I(y; x_1 | x_2) \leq \epsilon_{\text{info}}, \quad I(y; x_2 | x_1) \leq \epsilon_{\text{info}}. \tag{S1}$$

This is the foundational assumption underpinning self-supervised JE architectures (Shwartz-Ziv & LeCun, 2023). Several attempts have been made to relax this assumption (Kahana & Hoshen, 2022; Wang et al., 2022), although these rely on reconstructing the input image, which results in learning redundant features such as precise landmark locations and colours (Balestriero & LeCun, 2024), and consequently producing worse representations for downstream tasks in general. Thus, designing an architectural framework for relaxing the multi-view assumption is an open problem (Shwartz-Ziv & LeCun, 2023).

While the multi-view assumption is not especially restrictive for standard, multi-view approaches, these approaches become problematic in multimodal setting. This is because information not shared

between inputs can be ignored, degrading performance (Farndale et al., 2023b). Consequently, information in inputs cannot be divided as neatly into superfluous and predictive information based on being shared between inputs. From Equation S1 we can see that any information not shared between views is superfluous under mutual-information maximisation, hence additional modalities may stop the model learning useful unshared features. This is seen in ImageBind (Girdhar et al., 2023). Despite the emergence of alignment between modalities with no paired training data, the performance of the model is considerably worse than standard approaches. Nevertheless, aligning multiple inputs by mapping into shared latent spaces remains a promising approach for finding emergent alignment between feature sets.

## S1.3 MINIMAL SUFFICIENT REPRESENTATION

Representation learning is the task of finding a mapping $f$ from an input $x$ to a *representation* $z$ which is informative about some feature(s) of the data $y$. A *sufficient* representation of $x$ for $y$ is one which is as informative about $y$ as the input data itself, with no loss of information due to the encoding. Formally, this requires that the conditional mutual information $I(x; y|z) = 0$ (Wang et al., 2022).

Federici et al. 2020 show that conditional mutual information can be subdivided into predictive and superfluous information using the chain rule of information theory:

$$I(x; z) = \underbrace{I(x; z|y)}_{\text{superfluous information}} + \underbrace{I(y; z)}_{\text{predictive information}} . \tag{S2}$$

The *minimal sufficient representation* is the sufficient representation that most minimises the first term. In self-supervised learning, the label is assumed to be a priori unknown. Multi-view SSL methods therefore seek to minimise redundancy, with augmentations designed such that any information not shared between views will be an artefact of the augmentation, not helpful for a future predictive task. An input $x$ is augmented in two different way to give augmented inputs $x_1, x_2$, which are then mapped by the encoder $f$ to representations $z_1, z_2$. A projector $g$ then maps $z_1, z_2$ to embeddings $e_1, e_2$, on which the loss is applied. Reformulating Equation S2 for the unsupervised setting gives

$$I(x_1; e_1) = \underbrace{I(x_1; e_1|x_2)}_{\text{superfluous information}} + \underbrace{I(x_2; e_1)}_{\text{predictive information}} . \tag{S3}$$

In practice, it is not necessarily desirable for an encoder to learn a strictly sufficient representation, as this may only be possible by retaining an undesirable amount of superfluous information. As such, solutions typically seek to minimise the relaxed Lagrangian objective $\mathcal{L} = \mathcal{L}_{1 \to 2} + \mathcal{L}_{2 \to 1}$, where

$$\mathcal{L}_{1 \to 2}(\lambda_{12}) = I(x_1; e_1|x_2) - \lambda_{12} I(x_2; e_1), \tag{S4}$$

where $\lambda_{12}$ is the Lagrange multiplier induced by the optimisation problem, controlling the trade-off between the amount of superfluous information learned by the model and the amount of predictive information not learned, and $\mathcal{L}_{2 \to 1}$ is defined symmetrically (Federici et al., 2020).

### S1.3.1 TRIDENT AND MULTIPLE LATENT-SPACE MODELS

*TriDeNT* (Farndale et al., 2023b) is a model which was developed to address the restrictions of the multi-view assumption by adding an additional branch to the model architecture to create three simultaneous joint-embeddings. This allows the model to trade off learning simple features against complex features which receive a strong supervisory signal from the paired data. For a joint-embedding loss $\mathcal{L}$, TriDeNT minimises $\sum_{i,j \in \{1,2,*\}} \mathcal{L}(e_i, e_j)$. TriDeNT is effective for distilling information in a medical image setting, with significant performance improvements over baseline methods.

Despite TriDeNT's success with using knowledge distillation to guide the model to relevant, low-variance features, it is limited in scope by only being able to utilise one source of paired data. Additional joint-embeddings can overconstrain the model, leading to worse performance, as features shared between additional inputs but not present in the primary input receive a strong supervisory signal that does not relate to anything seen by the encoder. Alternatively, additional inputs may share few or no features, and cause the encoders to limit the variance of their learned features to avoid large

penalties from this mismatch. A natural approach might be to create joint embedding only between the primary branch and each additional branch in order to limit the effect of additional branches on each other, however, it has been shown that this still leads to emergent alignment between branches (Girdhar et al., 2023), meaning features not shared between branches are ignored.

## S2 EXTENDED DATASET DESCRIPTIONS

### S2.1 MNIST-CIFAR

The MNIST-CIFAR dataset is the standard test used to assess simplicity bias, and is designed to assess models' ability to learn complex features (Shah et al., 2020; Morwani et al., 2023; Tiwari & Shenoy, 2023). The dataset consists of digits from MNIST (LeCun et al., 1998) concatenated to images from the CIFAR10 dataset (Krizhevsky et al., 2009), as shown in Figure 2a. The MNIST images contain very simple features, and models typically achieve close to 100% accuracy on this test. In contrast, the features in CIFAR10 are more complex, and are not as easily learned. After training on the concatenated images, the level of simplicity bias can be assessed by randomising either the MNIST (denoted rMNIST) or CIFAR (denoted rCIFAR) component. The model's difference in performance between rNONE (no randomisation) and rMNIST determines the level of simplicity bias. If only the simple MNIST features have been learned, the performance will drop from near perfect to random guessing, while better accuracy indicates complex features have been learned from the CIFAR images. Conversely, if there is a performance drop on rCIFAR, the model has neglected some MNIST features and only learned the complex features.

Even if the encoder has learned both simple and complex features, the MNIST features will inevitably be more predictive and have greater variance. This means that the classifier will be biased towards the simpler MNIST features. A better test is therefore to train the classifier on the rMNIST dataset. This means that the classifier can only use the complex CIFAR features for classification, and directly assesses the complexity of the learned representations. In this setting we begin to see differences depending on the pretraining method.

### S2.2 EUROSAT

EuroSAT (Helber et al., 2019) is a well-established benchmark in geospatial image analysis. It consists of 27,000 $64 \times 64$px images at a resolution of 10 meters per pixel from the Sentinel-2 satellite, which is a part of the European Space Agency's Copernicus program. The images contain 13 spectral bands, which each correspond to a different region of the electromagnetic spectrum. These bands each contain different information about the land use and environmental properties of the region depicted in the image, such as cloud cover, soil moisture, and biomass estimation. The downstream task associated to EuroSAT is 10 class classification of the categories *Annual Crop*, *Forest*, *Herbaceous Vegetation*, *Highway*, *Industrial*, *Pasture*, *Permanent Crop*, *Residential*, *River*, *Sea Lake*.

### S2.3 MMEARTH

The MMEarth dataset (Nedungadi et al., 2024) is a large multimodal dataset containing data from 12 modalities for 1.2 million locations. We only utilise the Sentinel-2 data, which consists of 1.2 million $64 \times 64$px patches, with the same 13 spectral bands as EuroSAT. As there is no associated downstream task with MMEarth, we use EuroSAT to evaluate models trained on MMEarth.

### S2.4 NCT-CRC-100K

The NCT-CRC-100K dataset (Kather et al., 2019) is a widely used benchmark dataset in computational pathology. It features 100,000 $224 \times 224$px patches taken from 86 patients with colorectal cancer, approximately evenly split into 9 categories: *Adipose*, *Background*, *Debris*, *Lymphocytes*, *Mucus*, *Smooth Muscle*, *Normal Colon Mucosa*, *Cancer-Associated Stroma*, and *Colorectal Adenocarcinoma Epithelium*. There is also an associated test dataset containing 7180 patches from a separate 25 patients.

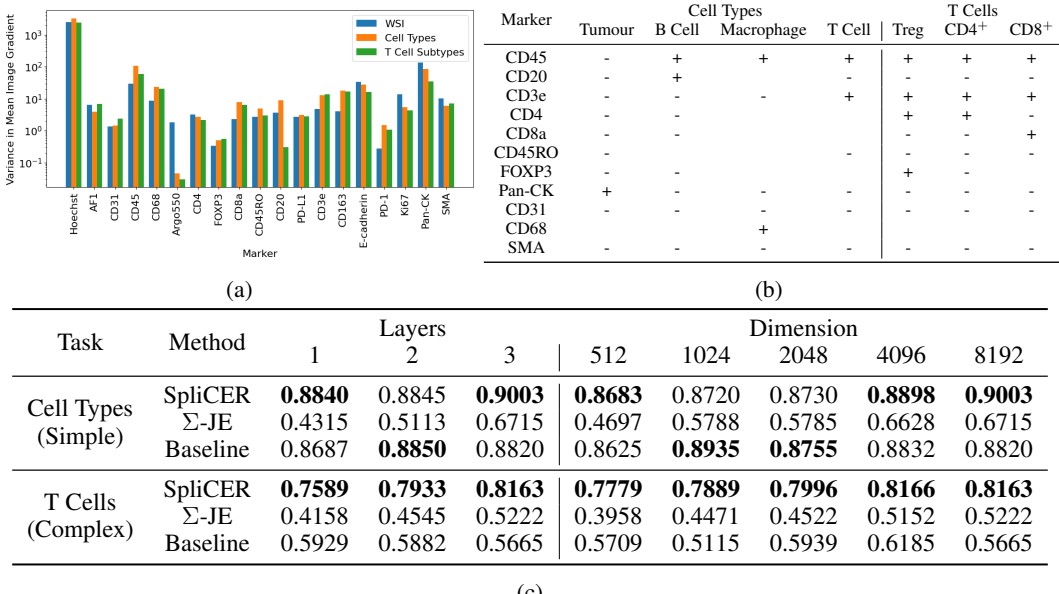

|        | Cell Types |        |            |        | T Cells |        |        |
|--------|------------|--------|------------|--------|---------|--------|--------|
| Marker | Tumour | B Cell | Macrophage | T Cell | Treg | CD4$^+$ | CD8$^+$ |
| CD45   | - | + | + | + | + | + | + |
| CD20   | - | + |   | - | - | - | - |
| CD3e   | - | - | - | + | + | + | + |
| CD4    | - | - |   |   | + | + | - |
| CD8a   | - | - |   |   | - | - | + |
| CD45RO | - |   |   | - | - | - | - |
| FOXP3  | - | - |   |   | + | - |   |
| Pan-CK | + | - | - | - | - | - | - |
| CD31   | - | - | - | - | - | - | - |
| CD68   |   |   | + |   |   |   |   |
| SMA    | - | - | - | - | - | - | - |

(a)                    (b)

| Task | Method | Layers | | | Dimension | | | | |
|------|--------|--------|--------|--------|--------|--------|--------|--------|--------|
|      |        | 1 | 2 | 3 | 512 | 1024 | 2048 | 4096 | 8192 |
| Cell Types (Simple) | SpliCER | **0.8840** | 0.8845 | **0.9003** | **0.8683** | 0.8720 | 0.8730 | **0.8898** | **0.9003** |
|      | Σ-JE | 0.4315 | 0.5113 | 0.6715 | 0.4697 | 0.5788 | 0.5785 | 0.6628 | 0.6715 |
|      | Baseline | 0.8687 | **0.8850** | 0.8820 | 0.8625 | **0.8935** | **0.8755** | 0.8832 | 0.8820 |
| T Cells (Complex) | SpliCER | **0.7589** | **0.7933** | **0.8163** | **0.7779** | **0.7889** | **0.7996** | **0.8166** | **0.8163** |
|      | Σ-JE | 0.4158 | 0.4545 | 0.5222 | 0.3958 | 0.4471 | 0.4522 | 0.5152 | 0.5222 |
|      | Baseline | 0.5929 | 0.5882 | 0.5665 | 0.5709 | 0.5115 | 0.5939 | 0.6185 | 0.5665 |

(c)

Figure S1: (a) Average intra-image variance of channels in Orion-CRC subsets (b) Gating strategy used to create tasks for downstream analysis (c) Projector ablations on Orion-CRC

## S2.5 CAMELYON

To assess the robustness of the models trained on NCT-CRC-100K to distribution shift, we use the WILDS Camelyon17 dataset (Koh et al., 2021). This is a variant of the original Camelyon17 (Bandi et al., 2018) dataset, which features 1400 whole slide images of H&E stained lymph node sections. The dataset is constructed to assess robustness to distribution shift, with slides from three hospitals used for training, and slides from a different hospital used for testing. The dataset features $96 \times 96$px patches with a train/test split of 179,394/146,722. We resize the patches to $224 \times 224$px for consistency with NCT-CRC-100K. The evaluation task associated with Camelyon is binary classification of the presence of metastasis.

## S2.6 ORION-CRC

To the best of our knowledge, there are no existing benchmarks for machine learning on multiplex images. We therefore construct two datasets by subsampling the publicly available Orion-CRC dataset (Lin et al., 2023). Orion-CRC contains 41 pathology slides, images with both H&E staining and 17-plex multiplex immunofluorescence. The markers used were *Hoechst*, *CD31*, *CD45*, *CD68*, *CD4*, *FOXP3*, *CD8α*, *CD45RO*, *CD20*, *PD-L1*, *CD3e*, *CD163*, *E-cadherin*, *PD-1*, *Ki67*, *Pan-CK*, and *SMA*, and the images contain an autofluorescence channel (AF1) and a negative control (Argo550).

Using the segmentation mask provided for the CRC02 slide, we randomly subsampled 1% of all cells to create a dataset of 12,606 cells, which were extracted in $64 \times 64$px patches. We then created two zero-shot downstream evaluation tasks from CRC01, following the same protocol but randomly selecting 2000 cells from each desired cell type, split evenly into train and test sets. Cells were defined by the gating strategy in Figure S1b For the *Cell Types* task, this was *Tumour*, *B Cells*, *Macrophages* and *T Cells*. This task requires only simple features, as cells have very different marker profiles and are mostly morphologically distinct (B and T cells are less morphologically distinct than the others but express different markers). For the *T Cells* task, we randomly selected *CD4$^+$*, *CD8$^+$*, and *regulatory T cells (Tregs)* as our cell types, which are all T cell subsets. This requires more complex features to be learned, as these channels have lower variance and are morphologically indistinguishable.

| Method | Cell Types (Simple) | T Cells (Complex) | Method | Cell Types (Simple) | T Cells (Complex) |
|---|---|---|---|---|---|
| SpliCER | 0.9003 | **0.8163** | Baseline | 0.8820 | 0.5665 |
| Multiple Encoders | 0.8928 | 0.8029 | + additional branches | 0.6578 | 0.5545 |
| Double Hyperstack Branch | **0.9035** | 0.8116 | + distinct projectors | 0.6715 | 0.5222 |
| Baseline + chunking | 0.8827 | 0.6069 | + chunking (SpliCER) | **0.9003** | **0.8163** |
| (a) | | | (b) | | |

Figure S2: (a) Ablations on Orion-CRC tasks showing possible adaptations to SpliCER (b) Ablations on Orion-CRC tasks breaking down each element of SpliCER

To analyse which channels models are most likely to focus on, we analyse the variance in the image gradients of each channel. Hoechst has the largest variance by a significant margin, whereas some stains, such as FOXP3, can have three orders of magnitude less variance. This means that models are likely to ignore features in FOXP3, which are uncommon and therefore can incur large penalties in the pretraining loss. In contrast, features in Hoechst will be readily learned. We also observe large differences between intensities of different subsets of cells. For example, there is much greater variance among T cells than among all cells in the sample, as FOXP3 is a marker for Tregs. In contrast, there is considerably less variance in CD20 and pan-CK, as T cells do not express the these proteins, so any expression is likely to be either off-target staining or a T cell being adjacent to a tumour cell or B cell.

## S2.7 MedMNIST

Inspired by the MNIST handwritten digits dataset (LeCun et al., 1998), MedMNIST (Yang et al., 2023) is a collection of 18 standardised biomedical datasets which have been preprocessed to ensure ease of use. The images are available in 28, 64, 128, and 224px sizes, and each dataset has one or more associated classification tasks. In this work we use both ChestMNIST and PneumoniaMNIST, which feature chest radiographs from the NIH-ChestXray14 dataset (Wang et al., 2017) and a dataset from Kermany *et al.*(Kermany et al., 2018).

ChestMNIST features 112,120 frontal-view X-rays from 30,805 patients, split into train, validation, and test sets of size 78468, 11,219, ad 22,433 respectively. PneumoniaMNIST features 5856 frontal-view chest X-rays, with train/validation/test splits of 4708/524/624 respectively.

## S3 Training and Implementation Details

Models were trained for 100 epochs with a batch size of 256. The backbone encoder was a ResNet-18 (He et al., 2016) for all tasks except the histology segmentation training, where ResNet-50 was used. Primary models were always trained from random initialisation and were appended with three layer projection heads with layer size 8192, batch normalisation between layers, ReLU activations, and a linear final layer. Figure S1c shows the results are robust to this choice. For SpliCER, the primary projection head was split into evenly distributed chunks, and the projection head was scaled accordingly by the number of channels. For example, there are 19 channels in the Orion-CRC dataset, so each projection head had output dimension 431. In the Orion-CRC and MMEarth examples, a single paired encoder was used for all single-channel inputs – two in total. For the MNIST-CIFAR and histology segmentation examples, a separate encoder was used for each branch, giving three encoders in total. An Adam optimiser (Kingma & Ba, 2014) with a warmup cosine learning rate was used, warming up the learning rate from 0 to $10^{-4}$ over the first 10% of epochs. There were no differences in the hyperparameters used for SpliCER and any baselines.

Two loss functions were tested: VICReg (Bardes et al., 2022) and SimCLR (Chen et al., 2020). VICReg is a non-contrastive loss which contains terms to regularise the variance and covariance of each embedding, and enforcing pairwise invariance between embeddings. SimCLR is a contrastive method using the InfoNCE loss (Oord et al., 2018), which is also used in methods such as CLIP (Radford et al., 2021). For VICReg, we used the standard parameters $\lambda = \mu = 25, \nu = 1$, and for SimCLR we used the temperature $\tau = 0.5$.

Table S1: Effect of Normalisation

| Task | Method | None | Normalise-only | Clip & Normalise |
|------|--------|------|----------------|------------------|
| Cell Types (Simple) | SpliCER | **0.9003** | **0.7550** | 0.6710 |
| | Σ-JE | 0.6715 | 0.6242 | 0.6018 |
| | Baseline | 0.8820 | 0.5940 | **0.7285** |
| T Cells (Complex) | SpliCER | **0.8163** | 0.3558 | **0.6789** |
| | Σ-JE | 0.5222 | 0.3525 | 0.4812 |
| | Baseline | 0.5665 | **0.3598** | 0.5582 |

Table S2: Evidence of usefulness of background features from the histology segmentation tasks

| Loss | Method | NCT | Camelyon |
|------|--------|-----|----------|
| VICReg | SpliCER | **0.9372** | **0.8230** |
| | TriDeNT (Nuclei Paired) | 0.9209 | 0.7763 |
| | Baseline (Nuclei Paired) | 0.8979 | 0.8013 |
| | Baseline | 0.8855 | 0.6822 |

For downstream tasks, encoder weights were frozen and a linear classifier with softmax activation was used to perform classification. Classifiers were trained for 100 epochs with data augmented with the same regime used for pretraining. All metrics reported are mean per-class accuracy.

## S4 ABLATIONS

### S4.1 NORMALISATION AND CLIPPING

We assess the effect of normalisation to demonstrate that simply normalising the inputs does not address feature variance. We employ two different normalisation schemes: normalisation, and normalisation with clipping. Multiplexed immunofluorescence typically have a long tail and background noise, so a standard approach is to clip the bottom and top 5% of values in each channel before normalising to zero mean and unit variance (Wölflein et al., 2023).

We find that the improved performance of SpliCER cannot be explained by implicit normalisation. In Table S1, we demonstrate that normalisation of the data only reduces performance, and SpliCER on unnormalised data consistently outperforms the other methods. We postulate than normalisation is harmful due to condensing the majority of values to a very small interval, as these images are dominated by a small number of large values.

### S4.2 ALTERNATIVE APPROACHES

In Figure S2 we investigate alternative design choices which could be made in SpliCER. We observe marginally worse performance using a separate encoder for each branch and using one encoder for all branches (in both cases there is a separate encoder for the primary branch). This could be due to a single shared encoder essentially seeing $19\times$ as many training examples. Intuitively, there could be scenarios where it was more important to use different encoders, such as when we have prior knowledge that different branches contain very different input images, or if there are different input dimensions.

We also experiment with using a second hyperstack, following TriDeNT (Farndale et al., 2023b). This does not appear to significantly affect the performance. This is likely because all of the information in the hyperstack is also present in at least one branch of the paired data, so there are no features which could be neglected by not being paired. We find that the embedding chunking mechanism alone is insufficient to achieve the performance gains of SpliCER. If there are not multiple branches, the chunking is redundant, and is essentially equivalent to halving the size of the projector, as the remaining half only receives signal from the variance and covariance loss terms.

Table S3: MNIST-CIFAR results for original classes (0: car, 1: truck) and for new, out-of-distribution classes (0: bird, 1: cat)

| Method | Original Classes | | | New Classes | | |
| | rNONE | rMNIST | rCIFAR | rNONE | rMNIST | rCIFAR |
|---|---|---|---|---|---|---|
| SpliCER | 0.9980 | 0.6840 | 0.9974 | 0.9974 | 0.6638 | 0.9980 |
| $\Sigma$-JE | 0.9990 | 0.7485 | 0.9990 | 0.9990 | 0.5855 | 0.9985 |
| Baseline | 0.9969 | 0.6275 | 0.9964 | 0.9959 | 0.6265 | 0.9964 |

In Figure S2b, we break down SpliCER into its constituent parts to assess the impact of each. We see that naïvely adding branches makes the downstream performance considerably worse. This is not significantly affected by adding distinct projectors, but the addition of chunking to create SpliCER achieves significant performance gains on the complex T Cells task, and restores performance on the simple Cell Types task.

Finally, in Table S2, we confirm that the background provides useful information in the NCT and Camelyon tasks. We train both a baseline VICReg model and a TriDeNT model with just the nuclei as paired data. This improves performance compared to the unpaired baseline, but fails to reach the performance of SpliCER. This indicates that by incorporating features from the background SpliCER has a better set of learned features for these downstream tasks.

## S5 ADDITIONAL ANALYSIS

### S5.1 TRIDENT OPTIMISATION PROBLEM

TriDeNT (Farndale et al., 2023b) has been proposed as a method to balance learning features which are strong in either the primary or paired inputs. This allows the model to use the paired data only so far as it is useful, and neglect it if not. Concretely, the optimisation problem becomes

$$
\begin{aligned}
\mathcal{L} &= \sum_{\substack{i,j \in \{1,2,*\} \\ i \neq j}} \mathcal{L}_{i \rightarrow j}(\lambda_{ij}) \\
&= \sum_{\substack{i,j \in \{1,2,*\} \\ i \neq j}} I(x_i; e_i | x_j) - \lambda_{ij} I(x_j; e_i).
\end{aligned}
\tag{S5}
$$

The increased degrees of freedom allow $\lambda_{ij}$ to be implicitly adjusted to cater to the paired information, such that information which may be considered superfluous in the unimodal formulation can now be learned if it is shared between the primary and paired data. This mitigates the effects of strong augmentation on the features learned, as features which would otherwise be lost to augmentation can now have a strong supervisory signal from the paired information. The loss function for TriDeNT does not achieve this optimally, as the optimisation process still penalises superfluous information between primary views, even if the same information is not superfluous for the primary/paired terms, and vice-versa.

### S5.2 MAPPING INPUTS TO ONE SHARED LATENT SPACE OVERCONSTRAINS MODELS

We note that there is a significant difference in the performance of $\Sigma$-JE between the MNIST-CIFAR and the other tasks. This can be understood in the context of the multi-view assumption, Equation S1. There is no restriction on models' ability to learn any features shared between all branches, as these are present in all views. Features of the primary input which are not shared between all branches inherently must be shared with at least one branch. When there are two paired branches, as in MNIST-CIFAR, the supervisory signal for each feature is half coming from the branch with the feature, and half coming from the branch without the feature. Therefore, the supervisory signal for that feature to the main encoder can still be reasonably strong. In contrast, if there are a large number of branches without the feature (or a strongly correlated feature), the signal is considerably diluted by unrelated features or collapsed dimensions, making the feature difficult to learn unless its underlying signal is very strong. This is discussed further in Section S5.3

Table S4: Full MNIST-CIFAR Results

| Loss | Paired Image | Method | rNONE (All) | | | rMNIST (Complex) | | | rCIFAR (Simple) | | |
|---|---|---|---|---|---|---|---|---|---|---|---|
| | | | rNONE | rMNIST | rCIFAR | rNONE | rMNIST | rCIFAR | rNONE | rMNIST | rCIFAR |
| VICReg | NONE | TriDeNT | 0.9954 | 0.5165 | 0.9939 | 0.6765 | 0.6230 | 0.5434 | 0.9929 | 0.5160 | 0.9939 |
| | | Baseline | 0.9908 | 0.5160 | 0.9908 | 0.6061 | 0.6410 | 0.4673 | 0.9918 | 0.5150 | 0.9918 |
| | MNIST | SpliCER | 0.9974 | 0.5150 | 0.9985 | 0.6459 | 0.6425 | 0.5066 | 0.9974 | 0.5150 | 0.9974 |
| | | TriDeNT | 0.9985 | 0.5160 | 0.9985 | 0.3668 | 0.4990 | 0.3449 | 0.9985 | 0.5160 | 0.9980 |
| | | Baseline | 0.9980 | 0.5160 | 0.9980 | 0.6245 | 0.5270 | 0.6071 | 0.9985 | 0.5165 | 0.9980 |
| | CIFAR | SpliCER | 0.9888 | 0.5145 | 0.9878 | 0.7199 | 0.6840 | 0.5235 | 0.9867 | 0.5150 | 0.9852 |
| | | TriDeNT | 0.8699 | 0.6155 | 0.7577 | 0.6776 | 0.6645 | 0.5010 | 0.8291 | 0.5400 | 0.8041 |
| | | Baseline | 0.7372 | 0.6745 | 0.5847 | 0.6883 | 0.6830 | 0.4827 | 0.6719 | 0.5020 | 0.7020 |
| | BOTH | SpliCER | 0.9964 | 0.5155 | 0.9964 | 0.6781 | 0.6675 | 0.4893 | 0.9969 | 0.5170 | 0.9969 |
| | | Σ-JE | 0.9934 | 0.4855 | 0.9929 | 0.6071 | 0.6530 | 0.4510 | 0.9934 | 0.4855 | 0.9934 |
| SimCLR | NONE | TriDeNT | 0.9974 | 0.5165 | 0.9974 | 0.6515 | 0.6390 | 0.5158 | 0.9974 | 0.5165 | 0.9969 |
| | | Baseline | 0.9974 | 0.5160 | 0.9964 | 0.6301 | 0.6260 | 0.5179 | 0.9974 | 0.5170 | 0.9964 |
| | MNIST | SpliCER | 0.9959 | 0.5175 | 0.9944 | 0.6224 | 0.6100 | 0.5036 | 0.9964 | 0.5170 | 0.9949 |
| | | TriDeNT | 0.9980 | 0.5155 | 0.9985 | 0.5776 | 0.5670 | 0.4995 | 0.9985 | 0.5160 | 0.9985 |
| | | Baseline | 0.9980 | 0.5155 | 0.9974 | 0.5643 | 0.5875 | 0.4852 | 0.9980 | 0.5155 | 0.9974 |
| | CIFAR | SpliCER | 0.9908 | 0.5140 | 0.9893 | 0.7128 | 0.6940 | 0.5117 | 0.9903 | 0.5130 | 0.9888 |
| | | TriDeNT | 0.9408 | 0.5330 | 0.9250 | 0.7097 | 0.6760 | 0.5255 | 0.9066 | 0.5020 | 0.9219 |
| | | Baseline | 0.8224 | 0.5950 | 0.7362 | 0.7005 | 0.6880 | 0.5158 | 0.7566 | 0.5030 | 0.7561 |
| | BOTH | SpliCER | 0.9964 | 0.5165 | 0.9949 | 0.6469 | 0.6735 | 0.4781 | 0.9969 | 0.5165 | 0.9964 |
| | | Σ-JE | 0.9959 | 0.4840 | 0.9964 | 0.6592 | 0.6840 | 0.4883 | 0.9969 | 0.4845 | 0.9959 |

### S5.3 USE-CASE FOR MAPPING ALL CHANNELS INTO A SHARED LATENT SPACE

There is a highly specific use-case for mapping all channels into a shared latent space: when a downstream task is known and all channels are strongly associated with the downstream label, there is some utility in training aligning all embeddings. As there is no requirement that the features learned by each branch are necessarily the same, there may be features that are highly correlated but bare no resemblance to each other, such as associating the shape of 0s and properties of cars. These can be mapped to the same embedding element, but provide no meaningful information about the other branch. In the extreme case, where the branches are not related to each other at all (such as MNIST-CIFAR), this causes collapse to any shared feature. For MNIST-CIFAR, the shared feature is the correspondence between e.g. 0 and car, or 1 and truck. In this case, we find that SSL training with unrandomised MNIST and CIFAR in a shared latent space causes collapse to a low-rank solution with very high correlation to the label. This essentially reduces the problem to supervised learning with feature regularisation, as the signal from the additional branch serves only to differentiate cars from trucks, or 0s from 1s, with no information about the features learned.

This approach could be helpful in very specific cases, however, it is not useful for general representation learning, as the resultant representations are not robust to domain shift, much like supervised learning. We show in Table S3 that changing the image classes used as CIFAR inputs considerably reduces the performance of the Σ-JE model, while SpliCER can generalise well. This is because SpliCER creates a supervisory signal that encourages the learning of generic image features, which Σ-JE lacks.

Table S4 shows the results for all models evaluated on all datasets, including those they were not trained on. We see that this reproduces the results of (Shah et al., 2020), where models without paired data collapse to random accuracy on rMNIST and achieve good results on rNONE and rCIFAR. Interestingly, we observe that models paired with CIFAR can perform on rMNIST even when trained on rNONE. This implies that they have neglected simple MNIST features in favour of learning CIFAR features, as MNIST features would likely be used by the classifier head when training on rNONE. We propose that SpliCER always performs poorly in this setting because it has learned both MNIST and CIFAR features.