# OpenReview forum: "Divide and Conquer Self-Supervised Learning for High-Content Imaging"
_ICLR.cc/2026/Conference — ICLR 2026 Conference Withdrawn Submission_

### Official Review · Reviewer_nv4K · 2025-10-30

**Soundness:** 3
**Presentation:** 3
**Contribution:** 3
**Rating:** 4
**Confidence:** 4

**Summary:**

The authors propose Split Component Embedding Registration (SpliCER), a training paradigm in which they split an image into a primary image and different image sections, enforcing learning of features of all sections. By doing so the method yields more diverse features as it forces the architecture to not just learn 'simple', predictive features, but to learn a diverse set of predictive features. Moreover, their method can be applied to various pre-training paradigms.
The authors evaluate SpliCER on various downstream datasets, showing performance improvement over their baselines.

**Strengths:**

The authors proposed method is creative and novel to me. Moreover, the proposed idea of enforcing feature diversity is orthogonal to the majority of current self-supervised learning paradigms, which is very nice.
Additionally, the authors communicate their idea with great clarity, which I commend them for.

**Weaknesses:**

While the authors claim their method to be superior to currently existing self-supervised learning paradigms, they only compare themselves to old pre-training methods of SimCLR and VICRegl. It would be interesting to have the authors evaluate if their method would be compatible with newer methods, in particular, iBOT. This is largely because I believe iBOT, which uses Masked Image Modeling, may show similar behavior: When doing a masked view of the image it may occlude many features (similar to the partial images in this paper), hence learn a diverse set of features as well. Generally I find the amount of baseline methods the authors pair their method with somewhat limited. Additional examples of their method, e.g. with MAEs would be great.

Moreover, I find the datasets the authors evaluate to be somewhat small in scale, and far from a more realistic setting. It would be interesting to see an ImageNet pre-training experiment to compare their performance when encountering prevalent larger-scale image datasets, or maybe a training on one of the larger (>10k Chest X-Ray datasets) to show the method translates.

**Justification**:
While I like the method, the current evaluation lacks baseline pre-training methods and larger-scale datasets to allow judging if the method translates to more real-world pre-training settings and actually works. In particular, an experiment on ImageNet-1k, would be nice to compare how the method translates to larger datasets, and the inclusion of another SSL baseline (iBOT) would be amazing to judge how their method synergizes with more state-of-the-art pre-training methods. Should the latter be provided I will raise my score.

**Questions:**

Q1: Have you tried combining SpliCER with MAEs or the iBOT paradigm?
Q2: Have you evaluated your pre-training method on dense downstream tasks?

---

### Official Review · Reviewer_JzWG · 2025-10-30

**Soundness:** 1
**Presentation:** 1
**Contribution:** 3
**Rating:** 0
**Confidence:** 4

**Summary:**

This work introduces SpliCER, a novel self-supervised method for high-content imaging. It is versatile in terms of the usage, from cell images, through CT scans, histology slides, up to geospatial data. It can be also combined with any ssl techniques, or wth at least contrastive learning techniques. Experiments are made on MNIST, CIFAR, Camelyon, Spatial Proteomics, Geospatial datasets. There is also a formal analysis of the method. SpliCER is compared only to general baseline, not other SSL methods for those modalities.

**Strengths:**

The work tackles an important research questions and problems.

It aims to show the universality of the approach, from cell imaging to geospatial data.

I do believe that the work is making a good contribution to the field, provides novelty and is significant, however I am making here a bit of trust in authors rather than being truly convinced.

**Weaknesses:**

Lots of opinions and general statements that are not backed by the literature, e.g. "A prime example is multiplex imaging: information-dense images containing more than three channels, each with a distinct meaning and significant variation in the complexity of the features in each channel. Despite their prevalence, multiplex images are generally understudied in computer vision, and largely lack dedicated methods to extract key information from them." Especially that there are methods for such imaging [1, 2, 3, 4, 5, 6, 7, 8] and there exists even benchmarks dedicated to those modalities [9].

Introduction of the work does not convey clearly what are the exact contributions of the work and how does this work address limitations of other works in HCI.

Figure 1 is referenced on page 4 while being presented on page 2. Also it is unclear, tiny font, hard to read. Figure 1a is not referenced in the text at all. I do not understand what is presented in Figure 1b, there is no description of details in the caption.

Regarding the related work spatial proteomics is not the only one type of a method to obtain such images in medicine. This can be also Cell Painting images [10], High Content Screening [11], Spatial Transcriptomics [12], or multiplexed Immunofluorescence Images [13].

"Despite the difficulty in creating manually labelled datasets for supervised learning", again that's not a fact backed up by the literature, rather an opinion. Some auxiliary task allowing supervised learning can be defined as in [1, 5, 8].

Line 194, what is Eq. S4?

In general, vague and unclear description of the method itself.

Treating each channel separately is not a novel concept [14]. Better discussion how it differ in SPLICER would be beneficial.

There is variety of methods for high content imaging and none of them is used as a baseline. Experiments are poor and rather simple (CIFAR / MNIST).

The manuscript is not well organized, theoretical analysis is after experiments. In general the work is very chaotic in terms of description.

[1] Caicedo, Juan C., et al. "Weakly supervised learning of single-cell feature embeddings." Proceedings of the IEEE Conference on Computer Vision and Pattern Recognition. 2018.

[2] Borowa, Adriana, et al. "Weakly-supervised cell classification for effective high content screening." International Conference on Computational Science. Cham: Springer International Publishing, 2022.

[3] Kraus, Oren, et al. "Masked autoencoders for microscopy are scalable learners of cellular biology." Proceedings of the IEEE/CVF Conference on Computer Vision and Pattern Recognition. 2024.

[4] Moshkov, Nikita, et al. "Learning representations for image-based profiling of perturbations." Nature communications 15.1 (2024): 1594.

[5] Sanchez-Fernandez, Ana, et al. "CLOOME: contrastive learning unlocks bioimaging databases for queries with chemical structures." Nature Communications 14.1 (2023): 7339.

[6] Lu, Alex X., et al. "Learning unsupervised feature representations for single cell microscopy images with paired cell inpainting." PLoS computational biology 15.9 (2019): e1007348.

[7] Phillips, Lawrence, and Rory Donovan-Maiye. "CellRep: A Multichannel Image Representation Learning Model." Proceedings of the Computer Vision and Pattern Recognition Conference. 2025.

[8] Liu, Gang, et al. "Learning molecular representation in a cell." ICLR 2025

[9] Borowa, Adriana, et al. "Decoding phenotypic screening: A comparative analysis of image representations." Computational and Structural Biotechnology Journal 23 (2024): 1181-1188.

[10] Cimini, Beth A., et al. "Optimizing the Cell Painting assay for image-based profiling." Nature protocols 18.7 (2023): 1981-2013.

[11] Singh, Shantanu, Anne E. Carpenter, and Auguste Genovesio. "Increasing the content of high-content screening: an overview." Journal of biomolecular screening 19.5 (2014): 640-650.

[12] Jaume, Guillaume, et al. "Hest-1k: A dataset for spatial transcriptomics and histology image analysis." Advances in Neural Information Processing Systems 37 (2024): 53798-53833.

[13] Yun, Sukwon, et al. "Mew: Multiplexed immunofluorescence image analysis through an efficient multiplex network." European Conference on Computer Vision. Cham: Springer Nature Switzerland, 2024.

[14] Doron, Michael, et al. "Unbiased single-cell morphology with self-supervised vision transformers." bioRxiv (2023).

**Questions:**

So to increase my score to 2, I would need to understand the contributions. So far I am not able to assess their novelty.

To increase the score to 4, I would need to see proper baselines and experimentations with the datasets included in other works. I would at least assumed that benchmark from [9] will be used or evaluation protocol from [3].

To increase my score to 6, I would expect the work to be clear, have images of good quality that I can understand with proper captions.

To increase my score to 8, I would expect much better contextualization of the work with regards to the literature presented in weaknesses section.

It may be the fact that the work is trying too be too universal and versatile to truly understand the contributions.

---

### Official Review · Reviewer_wBMd · 2025-11-01

**Soundness:** 2
**Presentation:** 2
**Contribution:** 2
**Rating:** 2
**Confidence:** 4

**Summary:**

This paper introduces SpliCER (Split Component Embedding Registration), a self-supervised learning architecture designed to address simplicity bias in high-content imaging. The core idea is to decompose images into components and align each component's embedding to a distinct chunk of the primary encoder's embedding. The authors claim this prevents models from learning only simple features while ignoring complex ones. The method is evaluated on MNIST-CIFAR, spatial proteomics (Orion-CRC), hyperspectral imaging (EuroSAT, MMEarth), and medical imaging (histopathology, chest X-rays).

**Strengths:**

Relevant Problem: The paper addresses a real issue in self-supervised learning—the tendency to learn simple features at the expense of complex ones, which is particularly relevant for scientific imaging applications.

Diverse Applications: The authors demonstrate their method across multiple domains (medical imaging, geospatial, proteomics), showing broad potential applicability.

**Weaknesses:**

### Major Issues:

1. **Limited Technical Novelty**:

   The core contribution—chunking embeddings and aligning different image components to different chunks—is incremental. The idea of learning from decomposed inputs and addressing feature learning biases has been extensively explored in prior work:

   - **Multi-view contrastive learning**: Tian et al. (ECCV 2020, "Contrastive Multiview Coding") and Bachman et al. (NeurIPS 2019, "Learning Representations by Maximizing Mutual Information Across Views") thoroughly explored learning from multiple views of the same input with information maximization objectives.

   - **Feature decorrelation and redundancy reduction**: Bardes et al. (ICLR 2022, "VICReg") and Zbontar et al. (ICML 2021, "Barlow Twins") addressed similar issues of learning diverse features through decorrelation mechanisms.

   - **Simplicity bias in self-supervised learning**: Shah et al. (NeurIPS 2020, "The Pitfalls of Simplicity Bias in Neural Networks") and Robinson et al. (NeurIPS 2021, "Can Contrastive Learning Avoid Shortcut Solutions?") systematically analyzed how models learn simple features over complex ones in SSL settings.

   The distinction from TriDeNT (Farndale et al., 2023b—already cited by authors) is primarily the embedding chunking mechanism, which is a relatively minor architectural modification rather than a fundamental methodological advance. The paper does not clearly articulate what prevents prior methods from achieving similar outcomes.

2. **Weak Theoretical Foundation**:

   a) **Section 5's gradient analysis is heuristic**: The analysis in Equations 4-6 assumes a "sufficiently sparse primary projector" (line 409) such that ∂ē_k/∂θ_{g,j} ≈ 0 for j ≠ k, but this assumption is never formally justified or empirically validated. The authors provide no:
      - Formal conditions under which this sparsity holds
      - Empirical measurements showing chunks are actually decoupled
      - Analysis of what happens when this assumption is violated

   b) **Information-theoretic claims lack rigor**:
      - Equation 1 presents an optimization objective based on mutual information (MI), but provides no discussion of how to estimate or optimize these MI terms in practice
      - The decomposition in Equation 3 into "superfluous" and "predictive" information is standard information bottleneck analysis
      - No formal proof that SpliCER avoids the stated issues
      - The claim that "features from X_k dominate the predictive information" leading to bias (lines 415-417) is stated without formal conditions

   c) **No sample complexity or convergence analysis**: The paper lacks any formal analysis of:
      - How many samples are needed for SpliCER to be effective
      - Convergence guarantees
      - Conditions under which the method provably learns complex features

3. **Problematic Experimental Design**:

   a) **MNIST-CIFAR is a weak benchmark**:
      - This is an artificially constructed dataset that doesn't reflect real high-content imaging scenarios
      - The "simple" vs. "complex" feature distinction is overly binary and doesn't capture the continuous spectrum of feature complexity in real scientific images
      - The baseline already achieves 64% on rMNIST (Table 2c), showing it learns both feature types; the improvement to 68.4% is marginal
      - A model trained on CIFAR alone achieves 72.6%, but SpliCER only reaches 68.4%—why not just train on CIFAR if that's the goal?

   b) **Insufficient baseline comparisons**:
      - **No comparison with modern SSL methods**: The paper only compares with VICReg (2022) and SimCLR (2020). Missing comparisons with:
        - **MAE** (He et al., CVPR 2022, "Masked Autoencoders Are Scalable Vision Learners")
        - **DINO** (Caron et al., ICCV 2021, "Emerging Properties in Self-Supervised Vision Transformers")
        - These methods may naturally learn diverse features through their architectures

      - **The Σ-JE baseline is non-standard**: This baseline is not established in literature, making it difficult to contextualize the contributions. How does SpliCER compare to simply training separate encoders for each component and concatenating representations?

      - **No ablation against redundancy reduction methods**: Papers like Barlow Twins (Zbontar et al., ICML 2021) explicitly address feature redundancy—how does SpliCER compare?



4. **Critical Missing Ablations**:

   a) **Chunking strategy**:
      - Only equal-size chunks are explored
      - What about non-uniform chunk allocation based on component importance?
      - How sensitive is performance to chunk size?
      - What happens with 2, 4, 8, or 16 chunks instead of n=number of components?

   b) **Number of components (n)**:
      - No systematic study of how performance scales with n
      - Does adding more components always help, or is there a point of diminishing returns?
      - What about computational cost as n increases?

   c) **Component selection**:
      - How should users choose which components to use?
      - What happens if uninformative components are included?
      - The paper mentions allocating "more features to informative channels" (line 481) but never demonstrates this experimentally

   d) **Comparison with simpler alternatives**:
      - Would simply augmenting each component separately achieve similar effects?
      - What about weighted loss functions that emphasize complex features?
      - Does the method work if you just train on the "complex" components directly?

   e) **Architecture choices**:
      - Why ResNets specifically? Would Vision Transformers behave differently?
      - How does projector architecture affect chunk specialization?
      - No ablation on projector depth, width, or activation functions



5. **Limited Experimental Scope**:

   a) **Small-scale experiments only**:
      - All experiments use relatively small ResNet models
      - No experiments with modern architectures (ViT, Swin Transformers)
      - No experiments at scale (ImageNet-scale pretraining)
      - Unclear how the method scales to foundation models

   b) **Domain-specific method comparisons missing**:
      - For spatial proteomics: No comparison with domain-specific methods
      - For medical imaging: No comparison with medical imaging SSL methods beyond generic VICReg/SimCLR
      - For hyperspectral: No comparison with remote sensing-specific methods

   c) **Insufficient evaluation metrics**:
      - Only downstream classification accuracy is reported
      - **No analysis of what chunks actually learn**: Are they truly specialized? Is there redundancy?
      - **No feature visualization**: What do different chunks capture?
      - Figure 3c shows correlation with marker intensities for Orion-CRC only—what about other datasets?
      - No analysis of feature diversity, rank, or representational geometry


### Minor Issues:

6. **Presentation and Clarity Issues**:


   a) **Notation inconsistency**:
      - Both z and e are used for representations/embeddings with unclear distinction
      - Sometimes "representation" and "embedding" are used interchangeably
      - The relationship between z_primary, ē, and ē_j needs clearer exposition



7. **Related Work Limitations**:

   The paper appropriately cites relevant work on simplicity bias (Shah et al., NeurIPS 2020; Geirhos et al., Nature MI 2020; Robinson et al., NeurIPS 2021), multi-view learning (Bachman et al., NeurIPS 2019; Radford et al., ICML 2021), and self-supervised learning (Bardes et al., ICLR 2022; Chen et al., ICML 2020). However:

   a) **Insufficient contextualization**: The related work doesn't clearly explain how SpliCER differs from multi-view SSL methods beyond the chunking mechanism

   b) **Missing recent developments**: No discussion of recent SSL advances like MAE (He et al., CVPR 2022), DINO (Caron et al., ICCV 2021), or their potential applicability to this problem

   c) **Limited discussion of alternatives**: The paper doesn't discuss why existing approaches to learning diverse features (e.g., Barlow Twins' redundancy reduction) are insufficient

**Questions:**

Theoretical justification: Can you provide formal theoretical analysis showing when/why SpliCER provably avoids simplicity bias? Under what conditions does the gradient decoupling actually occur?

Ablation studies: What happens with different numbers of chunks (e.g., 2, 4, 8, 16)? How sensitive is performance to the chunking strategy? What if chunks have different sizes based on component importance?


Baseline comparisons: Can you compare with:Modern SSL methods (MAE, DINO, DINO v2)? Feature disentanglement methods? Simply training separate encoders for each component and concatenating?


Decomposition analysis: How do you determine the optimal decomposition for a new dataset?
What happens if the decomposition is suboptimal?
Can you provide experiments with different decomposition strategies for the same dataset?


Learned representations: Can you visualize what each chunk learns?
How do chunks specialize on different features?
Is there any redundancy across chunks?


Scalability: How does SpliCER perform with larger models (ViT-L, ViT-H)?
What's the computational overhead compared to baseline SSL?
How does it scale with the number of components (n)?


Statistical significance: Can you provide:
Multi-seed experiments with error bars?
Statistical significance tests for the improvements claimed?
Analysis of variance across different random initializations?


Real-world applicability:
For the segmentation-based experiments, how sensitive is the method to segmentation errors?
What happens when components have vastly different information content?
Can this work without any prior knowledge about image structure?

---

### Official Review · Reviewer_N1Se · 2025-11-02

**Soundness:** 3
**Presentation:** 3
**Contribution:** 3
**Rating:** 4
**Confidence:** 4

**Summary:**

Self-supervised learning methods often favor simpler features, overlooking the subtle, complex patterns critical for scientific discovery. To address this, the authors propose SpliCER, a architecture that splits images into sections to guide the model in learning both simple and complex features without compromise. Compatible with any self-supervised loss, SpliCER demonstrates significant performance improvements in medical and geospatial imaging, providing a powerful new tool to uncover features that other methods may miss.

**Strengths:**

Cross-Domain Validation: Extensive experiments across synthetic (MNIST-CIFAR), medical (Orion-CRC, histopathology), and geospatial (EuroSAT, MMEarth) domains demonstrate broad applicability.
Methodological Rigor: Well-designed ablations with strong baselines (VICReg, SimCLR) and elegant simplicity bias quantification through controlled experiments.
Practical Impact: Addresses critical needs in scientific imaging where subtle features are crucial, with particularly strong results on fine-grained tasks and distribution shift scenarios.
Flexibility: Works with existing image structures (multiplex channels) or readily available segmentation tools (SAM, HoVer-Net).

Strong grounded theoretical motivation : The connection of breaking down embedding component-wise is well motivated and could be extremely beneficial.

**Weaknesses:**

1)Computational & Scalability Concerns
No quantification of memory usage, training time, or inference speed compared to standard SSL methods
Scalability Questions: How does computational overhead scale with number of components/chunks? What about GPU memory requirements for multiple encoders?
Practical Deployment: Real-world implementation costs could be prohibitive without understanding the computational trade-offs

2. Methodological Gaps
Hyperparameter Sensitivity: No systematic study of how chunk allocation affects performance - this is critical for practitioners
Architecture Dependencies: Unclear how the method performs across different backbone architectures beyond those tested
Component Definition Ambiguity: The paper doesn't provide clear guidelines for identifying meaningful "components" in new domains
Registration Mechanism Details: The embedding registration process lacks mathematical rigor - how exactly are embeddings "registered" to specific chunks?

3. Experimental Limitations
Limited Baseline Diversity: Missing comparisons to recent SSL methods that address feature learning biases https://arxiv.org/abs/2409.18316 (I dont expect the authors to experiment but perhaps address them in terms of methodology)

Dataset-Specific Tuning: Results may be optimized for the specific datasets used - generalization unclear
Statistical Significance: No error bars, confidence intervals, or statistical tests across multiple runs

Ablation Incompleteness: Missing ablations on key design choices (embedding dimension allocation, component weighting)

4. Theoretical Foundations
Feature Complexity Definition: Post-hoc labeling of features as "simple" vs "complex" without principled criteria. This remains a core criterion while motivated well makes me question do we need expert annotation for this.

Simplicity Bias Quantification: Beyond MNIST-CIFAR, no rigorous measurement of when/why simplicity bias occurs
Theoretical Guarantees: No analysis of when SpliCER is expected to work vs fail

5. Implementation

Implementation Details: Missing crucial details about training procedures, convergence criteria, hyperparameter selection
Component Extraction: Insufficient detail on how to apply segmentation tools effectively in practice

**Questions:**

1)How exactly do the paired encoders process components M and N? Are they shared or separate networks?
2)What's the relationship between primary embedding dimension and total chunk allocations?
3)How would one prevent mode collapse when multiple components learn similar features?
4)Can you provide the complete mathematical formulation for the registration mechanism?
5)What is Equation S4 referenced in the text?
6)How do you ensure gradient flow balance across all component encoders?

7)What are the exact training times and memory requirements vs baselines?
8)How does performance degrade with computational budget constraints?
9)What's the minimum viable computational setup for practical deployment?

10)How sensitive is performance to chunk size allocation strategies?
11)What happens when you have mismatched numbers of components across images? Given the proteomics data this seems quite common.
12)How do you determine optimal embedding dimension allocation per component?

13)How does performance vary across different backbone architectures (ResNet, ViT, etc.)?
14)How robust is the method to poor quality segmentation masks? For example SAM is known to be prone to OOD modalities so I am keen to understand this.

15)Can you provide quantitative metrics for measuring feature complexity beyond task performance?
16)How do you validate that "complex" features are actually being learned vs memorized?
17)What visualization or interpretation tools can demonstrate the method's effectiveness?

18)How does SpliCER compare to other bias mitigation techniques in SSL?
19)What about comparison to multi-task learning approaches that might achieve similar goals?
20)Can you provide head-to-head comparisons with TriDeNT on identical experimental settings?

---

### Note · Authors · 2025-11-14

I have read and agree with the venue's withdrawal policy on behalf of myself and my co-authors.